# ARE LARGE VISION-LANGUAGE MODELS ROBUST TO ADVERSARIAL VISUAL TRANSFORMATIONS?

## ABSTRACT

Large Vision-Language Models (LVLMs) have demonstrated remarkable capabilities across a wide range of multimodal understanding and reasoning tasks. However, recent research shows that LVLMs are susceptible to adversarial examples. Existing LVLM attackers either optimize the perturbations on the visual input or manipulate prompts to fool the LVLM models, requiring extensive design and engineering on these adversarial manipulations. While straightforward visual transformation can boost training generalization-ability, its potential risks to LVLMs in terms of safety and trustworthiness have been largely neglected. In this paper, we ask an intriguing question: *can simple yet easy-to-implement adversarial visual transformations be utilized to attack the LVLM models?* Motivated by this research gap and new attack setting, we propose the first comprehensive assessment of LVLMs' adversarial robustness to visual transformations by testing LVLMs' resilience to all possible transformation operations. Our empirical observations suggest that with the appropriate combination of the most harmful transformations, we can build transformation-based attacks more adversarial to the LVLM models. Moreover, adversarial learning of visual transformations is further introduced to adaptively apply the malicious impacts of all potentially harmful transformations to the raw images via gradient approximation for improving the attack effectiveness and imperceptibility. We hope that this study can provide deeper insights into the potential vulnerability of LVLMs to adversarial visual transformations.

## 1 INTRODUCTION

Large Vision-Language Models (LVLMs) have demonstrated exceptional abilities in various multimodal downstream tasks, such as text-to-image generation (Nichol et al., 2021; Ramesh et al., 2022; Rombach et al., 2022), visual question-answering (Tsimpoukelli et al., 2021; Li et al., 2023; Alayrac et al., 2022), and *etc.*. Despite their remarkable capabilities, the increased complexity and deployment of LVLMs have also exposed them to various security threats and vulnerabilities. Current studies (Luo et al., 2024; Zhao et al., 2024) have shown that LVLMs are vulnerable to adversarial examples. These examples are typically developed by adding subtle yet invisible perturbations, but can significantly degrade the LVLMs' performance, posing critical safety issues.

Existing LVLM attackers (Bailey et al., 2023; Dong et al., 2023; Wang et al., 2023b; 2024b; Zhang et al., 2024; Lu et al., 2024; Luo et al., 2024; Tao et al., 2024; Zhao et al., 2024) generally craft perturbations to benign image/text inputs or manipulate visual/textual prompts for fooling the LVLM model (Fan et al., 2024; Liu et al., 2024b). As for the perturbation-based attacks (Qi et al., 2024; Luo et al., 2024; Bailey et al., 2023; Lu et al., 2024; Zhao et al., 2024; Dong et al., 2023), they require carefully noise-style designs with explicit loss constraints to ideally optimize the perturbations for integration with the original data. As for the structure-based attacks (Shayegani et al., 2023; Gao et al., 2024b; Bagdasaryan et al., 2023; Chen et al., 2023; Wu et al., 2023), they require extensive manual engineering on the malicious prompts with additional tools like text-to-image models (Shayegani et al., 2023) to guide the model to conduct unsafe behaviors. Although the above two types of methods achieve significant attack performance, they heavily rely on the abundant attack pattern/flow designs without making an in-depth investigation into the self-robustness of LVLMs.

Considering that visual transformation generally serves as an essential augmentation tool (He et al., 2016; Krizhevsky et al., 2012; Simonyan & Zisserman, 2014) to improve the model's robustness and

generalization-ability, we propose to assess the self-robustness of LVLM by posing a key question: *can adversarial visual transformations compromise the reasoning ability and textual output semantics of LVLMs?* Investigating this question is crucial to revealing the vulnerability of LVLM to these simple yet easy-to-implement visual transformations and providing potential defense insight. From the perspective of an attacker, adversarial examples designed with solely visual transformations can mislead LVLMs to generate malicious outputs without relying on redundant attack designs or using additional model tools. From a defensive standpoint, the potential to obfuscate image information can reflect the LVLM's weakness to the unseen transformations, providing a promising solution for robust LVLM training/fine-tuning with suitable augmentation strategies.

To address this research gap, in this paper, we propose a comprehensive assessment of LVLMs' adversarial robustness to visual transformations by evaluating LVLMs across three key dimensions: *(i)* LVLM's resilience against different individual transformations, *(ii)* the degree of harmfulness of different transformation combinations, and *(iii)* the adversarial performance on multiple LVLM models and datasets. Furthermore, we undertake a comprehensive exploration to manually identify an optimal combination of the most harmful transformations for attacking LVLMs, examining the impacts of different visual transformations across datasets and models. Moreover, to design more superior adversarial transformations, we further introduce adversarial learning strategies to transfer the transformation impacts via a gradient approximation problem to perturb the raw images by adaptively mimicking the unknown but most harmful transformation combinations for improving the attack effectiveness and imperceptibility. Our key contributions are outlined as follows:

- We rigorously assess LVLMs' robustness to a broad range of visual transformations, aiming to reveal LVLMs' vulnerability to these transformation operations and illuminate the path toward effective transformation-based adversarial attacks.

- Our empirical observations show that different transformations share diverse harmfulness degrees on different LVLMs' models and datasets. With appropriate combination of the most harmful transformations, we can manually build transformation-based attacks that are more adversarial to the LVLM models.

- Our study also reveals that this transformation-based attack further benefits from existing adversarial learning algorithms and gradient approximation techniques aimed at enhancing security and truthfulness.

- These insights are derived from extensive experiments on different LVLM models and multiple datasets. Results suggest that the potential of simple and easy-to-implement adversarial visual transformations can be effectively harnessed to fool the LVLMs.

## 2 RELATED WORK

**Adversarial Robustness of LVLMs.** Despite achieving impressive performance, LVLMs still face issues of adversarial robustness due to their architecture based on deep neural networks (Szegedy et al., 2013). Multiple primary attempts have been conducted to study the robustness of LVLMs from different aspects. Most LVLM attacks follow a perturbation-based approach (Qi et al., 2024; Luo et al., 2024; Bailey et al., 2023; Lu et al., 2024; Zhao et al., 2024; Dong et al., 2023; Wang et al., 2023b), which involves introducing adversarial perturbations into the input data, often in a way that is imperceptible to humans. These perturbations are designed to exploit the vulnerabilities in the model's processing of input data, causing the model to output incorrect or harmful responses. Different settings of white-box (Schlarmann & Hein, 2023; Cui et al., 2023; Luo et al., 2024; Gao et al., 2024b; Bailey et al., 2023; Gao et al., 2024a; Wang et al., 2024b), gray-box (Wang et al., 2024a; Dong et al., 2023; Zhao et al., 2024; Tu et al., 2023; Guo et al., 2024), and black-box (Zhang et al., 2024) requires different levels of access attackers have to the victim model. Instead of optimizing perturbations, structure-based attacks (Shayegani et al., 2023; Gao et al., 2024b; Bagdasaryan et al., 2023; Chen et al., 2023; Wu et al., 2023) are proposed to employ simple typography or text-to-image tools to manually design the multimodal inputs of LVLMs. These attacks involve transferring the harmfulness of text into images, using inducing textual prompts to direct LVLMs to focus on malicious content within the images, thereby circumventing safety checks to achieve the attack's aim. However, the above two types of methods severely rely on the abundant attack designs and engineering on adversarial manipulations. Our work tries to design attack in a more simple yet easy-to-implement transformation perspective.

**Visual Data Augmentations.** Data augmentations often transform (*e.g.*, flipping, rotation, cropping, *etc.*) the image during the training process for better generalization. Mixup (Zhang et al., 2017) interpolates two images and their labels to generate virtual samples for training with various transformations. Cutmix (Yun et al., 2019) pastes an image patch to the original patch and mixes the labels accordingly. AutoAugment (Cubuk et al., 2019) automatically searches for improved data augmentation policies (operations and parameters) on the dataset for better generalization, which has been widely adopted in deep learning. Unlike these data augmentation strategies, we aim to construct a set of diverse images by transforming the image using various transformations to assess the vulnerability of LVLMs and accordingly design transformation-based attacks.

## 3 How Do LVLMs Perform under Visual Transformations?

**Takeaways:** ❶ Visual transformations can effectively affect the textual output semantics of LVLMs. ❷ Block-level transformations of rotation, vertical flip, horizontal shift, and vertical shift are the most harmful individual transformations. ❸ By further appropriately combining different transformations, we can generate more harmful transformation operations to attack LVLMs.

### 3.1 Preparation for Visual Transformations

To evaluate the adversarial robustness of LVLMs against visual transformations, we first select multiple basic image transformation operations in both the spatial domain (Wang et al., 2023a) and spectral domain (Duan et al., 2021), then feed the transformed images into the LVLM for assessment. Specifically, the spatial transformations consist of 10 types, including Resize, Horizontal Flip, Vertical Flip, Rotate, Horizontal Shift, Vertical Shift, Scale, Add Noise, Dropout, and Color Jitter. The spectral transformation includes dropping frequency components. Each transformation is implemented in various types. Further, we also split the image uniformly into multiple patches with the same size, and perform basic transformations on each patch to get corresponding block-level transformations (AprilPyone & Kiya, 2021). There are 31 transformations in total. More details of these transformation operations and corresponding visualizations are illustrated in the Appendix A.1.

### 3.2 LVLM Models, Datasets, Metrics and Set-up

**LVLM Models.** We conduct our experiments on four popular popular open-source LVLM models, including LLaVA-1.5 (integrated with Vicuna-7B) (Liu et al., 2024a), MiniGPT-4 (integrated with Llama-2-7B-Chat) (Zhu et al., 2023), BLIP-2 (integrated with OPT-2.7b) (Li et al., 2023), and InstructBLIP (integrated with Vicuna-7B) (Dai et al., 2024).

**Datasets.** We evaluate the adversarial robustness on three multi-modal datasets for the image captioning, image classification, and VQA tasks. The datasets consist of both images and prompts. The images are collected from three datasets: VQAv2 (Goyal et al., 2017), SVIT (Zhao et al., 2023), and DALL-E (Ramesh et al., 2021). The prompts for image captioning and image classificaiton derive from the CroPA (Luo et al., 2024). The prompts for VQA come from the Anydoor (Lu et al., 2024).

**Evaluation Metrics.** To measure the semantic changes of the LVLM's output, we follow previous work (Zhao et al., 2024) to utilize the SentenceTransformer (Reimers & Gurevych, 2019) to generate embeddings of both adversarial and original outputs for calculating their cosine similarity. The lower similarity denotes the semantic change is large and the transformation is more adversarial.

**Implementation Details.** All experiments of this section are implemented on a single NVIDIA RTX 4090 24G GPU. In particular, we utilize 357 images from the VQAv2 dataset, 329 images from the SVIT dataset, and 200 images from the DALL-E dataset. The average running time for feeding a transformed image and a textual prompt into the LVLM and getting a response is about 4s, which is very efficient. The GPU memory occupied by LLaVA-1.5, MiniGPT-4, BLIP-2, and InstructBLIP models are approximately 15GB, 10GB, 7GB, and 17GB, respectively.

### 3.3 Evaluation Results

**Can Visual Transformations Affect the LVLM's Performance?** To investigate the harmfulness of different visual transformations, we assess their individual performance on four LVLM models

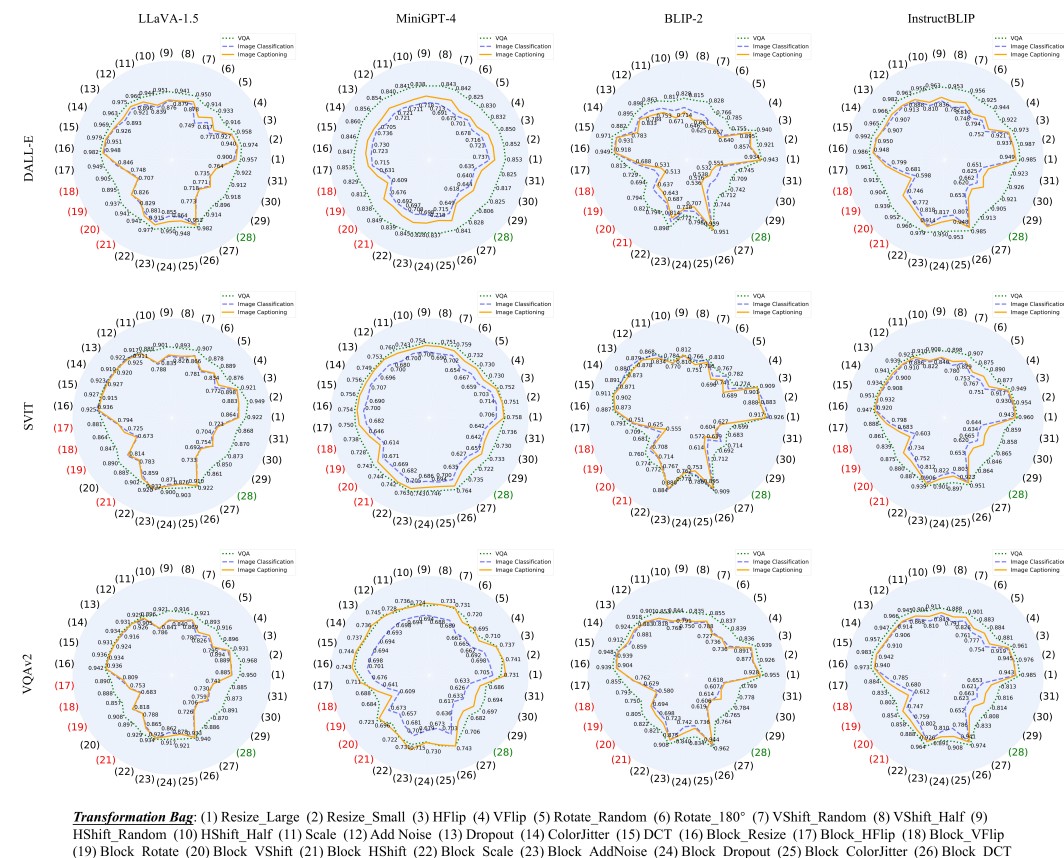

**Transformation Bag**: (1) Resize_Large (2) Resize_Small (3) HFlip (4) VFlip (5) Rotate_Random (6) Rotate_180° (7) VShift_Random (8) VShift_Half (9) HShift_Random (10) HShift_Half (11) Scale (12) Add_Noise (13) Dropout (14) ColorJitter (15) DCT (16) Block_Resize (17) Block_HFlip (18) Block_VFlip (19) Block_Rotate (20) Block_VShift (21) Block_HShift (22) Block_Scale (23) Block_AddNoise (24) Block_Dropout (25) Block_ColorJitter (26) Block_DCT (27) Block_Rotate_HShift (28) Block_Rotate_VFlip (29) Block_VFlip_HShift (30) Block_Rotate_VFlip_HShift (31) Block_Random_Combination

Figure 1: Evaluation results of our implemented 31 number of transformations on four LVLM models across three datasets with three tasks. Lower similarities (↓) indicate more harmful impacts. Red: the top-4 harmful transformations in (1)-(26); Green: the top-1 harmful transformation in (27)-(31).

in three tasks with three different datasets. As shown in Figure 1, each radar chart represents the evaluation of the adversarial robustness of all implemented transformations, where each point represents the semantic similarity between the adversarial output of a specific transformed input and the original output. The farther the point is from the edge, the stronger the adversarial effect of corresponding transformation operation. From this figure, we can conclude that:

*(i) All visual transformations can affect the output semantics of LVLMs.* Specifically, for a certain LVLM model, each transformation can degenerate the textual output performance and has a similar effect on this model across different datasets. We think this is due to the invariant self-robustness of the LVLM model. For example, the affected similarities of transformations (1)-(15) on LLaVA-1.5 for the image classification task on DALL-E dataset are 0.900, 0.940, 0.927, 0.771, 0.817, 0.749, 0.878, 0.879, 0.876, 0.839, 0.896, 0.921, 0.893, 0.926, 0.951 respectively. These transformations also achieve similar performance on LLaVA-1.5 for VQA and captioning tasks across datasets, with the lowest similarities of (1)-(15) reaching 0.914, 0.876, 0.893 and 0.807, 0.793, 0.792 on DALL-E, SVIT, and VQAv2 datasets respectively. Similar phenomena also occur in other LVLM models.

*(ii) Different transformations have different impacts on LVLMs.* Moreover, the block-level transformations (16)-(26) can make the LVLM's result more adversarial compared to their nonblock ones (1)-(15) due to their more complicated and diverse operations. For example, in the image captioning task on the DALL-E dataset, the semantic similarity of LLaVA-1.5 on Rotate_180°, VFlip, and Scale operations are 0.807, 0.814, and 0.920 respectively, while on corresponding block-level transformation, they achieve much lower 0.711, 0.757, and 0.903. This shows that block-level transformation has a more harmful impact on the robustness of the model.

In summary, the general visual transformations can effectively affect the performance of LVLMs, revealing the LVLM's vulnerability to potential visual transformations.

**Which Transformation is More Harmful?** To investigate the most harmful transformations for latter adversarial transformation designing, we provide a deep analysis according to Figure 1 as:

*(i) Rotation, vertical flip, horizontal shift, and vertical shift operations have more adversarial impacts than other transformations on LVLMs.* From this figure, we can find that the points of these four operations are always the farthest from the edge points among the basic transformations (1)-(15) in different tasks on different datasets. For example, for the image captioning task on the DALLE dataset, Rotate_180°, VFlip, HShift_Half, and VShift_Half transformations have the greatest impact on LLaVA-1.5, with the harmful results reaching 0.807, 0.814, 0.856, 0.880 respectively. However, transformations like DCT, Resize_Small, Add Noise and ColorJitter have lower harmful impacts on LLaVA-1.5, which only achieve 0.957, 0.943, 0.943, 0.931. The corresponding block-level transformations of these four operations also achieve the most harmful performance among (16)-(26). Therefore, in all basic transformations (1)-(26), block-level rotation, vertical flip, horizontal shift, and vertical shift, *i.e.*, transformation (18)(19)(20)(21), are the most harmful transformations.

*(ii) Further combining above transformations can achieve more harmful results.* In addition to exploring the performance on individual transformation, we also investigate whether transformation combination can further degenerate the performance of LVLM models. According to the performance of combined transformations (27)-(31) in the figure, we can find that the combined transformations are generally more harmful than the impacts of their contained individual transformations. In particular, applying both block-level rotation and block-level vertical flip to image input (*i.e.*, transformation (28)) can achieve the lowest semantic similarities among the four LVLMs.

In summary, block-level transformations of rotation, vertical flip, horizontal shift, and vertical shift are the most harmful individual transformations, and their further combination can achieve more harmful results. More analysis and textual output visualizations can be found in Appendix A.2.

## 4 How to Design A Superior Adversarial Visual Transformation Against LVLMs?

Enlightened by the above insights into the impacts of different visual transformations, we can manually construct the most harmful transformation combinations and apply them on the input images to fool the LVLM models. We further design an adversarial learning strategy to adaptively generate superior adversarial visual transformations for improving both the attack effectiveness and imperceptibility of adversarial samples in untargeted and targeted scenarios.

**Takeaways:** ❶ By manually enumerating and assessing different transformation combinations, we can construct and formulate much more harmful impacts than the general transformations in Section 3 (Comparison on averaged semantic similarity↓: 0.568 *vs.* 0.683). ❷ To further boost the efficiency and effectiveness, we can utilize the adversarial learning strategy to adaptively search for all potential harmful transformations and impose their adversarial impacts on the raw images to achieve the most adversarial performance while improving the imperceptibility of the disturbed images. ❸ Our developed adversarial transformations achieve significant performance in both challenging untargeted and targeted attack settings, demonstrating the great practicality and scalability.

### 4.1 Preliminary

**Evaluation Metrics.** We consider two metrics in our experiments, namely semantic similarity and attack success rate. For untargeted attacks, we utilize the SentenceTransformer (Reimers & Gurevych, 2019) to generate embeddings of both adversarial and original outputs for calculating their cosine similarity (the lower the better). For targeted attacks, we not only utilize the semantic similarity to measure the distance between adversarial output and target text (the larger the better), but also follow (Luo et al., 2024; Lu et al., 2024) to exploit success rates "ExactMatch" and "ConditionalContain" to assess the word-level overlap between adversarial output and target text.

**Implementation Details.** We utilize the same experimental resources and data following Section 3 to generate the adversarial images. In particular, we utilize the PGD algorithm (Madry et al., 2017) to optimize the adversarial perturbations with a maximum of $epoches = 500$. The perturbation size $\epsilon$ are set as $16/255$ and $32/255$, respectively. We set the number of transformed images for gradient calculation as $N = 20$, the momentum parameter as $\mu = 0.9$ and the step size as $\alpha = \epsilon/epoches$.

Figure 2: Illustration of our designed hybrid transformation-based attack, which manually constructs the most harmful transformation combination via enumeration (More details are in Appendix B.1).

## 4.2 MANUALLY CONSTRUCTING MOST HARMFUL COMBINATION OF TRANSFORMATIONS

**Designed Hybrid Transformation-based Attack.** Based on the observations in Section 3, we can manually construct the superior adversarial operation against LVLMs by appropriately combining the most harmful individual transformations. Specifically, since the block-level transformation is more harmful, we uniformly split each image into $3 \times 3$ patches and explore the vulnerability of each patch by separately enumerating transformation combinations among Rotate, VFlip, VShift, and HShift. In particular, for each patch, we are able to select one, two, three, or all four operations from these transformations to combine and apply, leading to 15 choices: (1) Rotate, (2) VFlip, (3) VShift, (4) HShift, (5) Rotate+VFlip, (6) Rotate+VShift, (7) Rotate+HShift, (8) VFlip+VShift, (9) VFlip+HShift, (10) VShift+HShift, (11) Rotate+VFlip+VShift, (12) Rotate+VFlip+HShift, (13) Rotate+VShift+HShift (14) VFlip+VShift+HShift, and (15) Rotate+VFlip+VShift+HShift. As shown in Figure 2, our hybrid transformation-based attack transforms each patch one by one in the default order to iteratively make the transformed image as harmful as possible. Starting from the first patch, we fix the remaining patches unchanged and perform the above 15 transformation operations in sequence to obtain the corresponding 15 transformed images. Then each transformed image is fed into the LVLM model individually with the same textual prompt to obtain the corresponding 15 adversarial answers. Next, we calculate the semantic similarities between these adversarial answers and the original answer, and select the operation with the lowest similarity score as the optimal (most harmful) transformation operation for this patch. By fixing the transformed patch 1, we repeat this process for patch 2 to further degenerate the LVLM's performance. After traversing all patches, we can transform images that pose a greater hazard than those described in Section 3.

**Evaluation and Discussion.** We evaluate the performance of our designed hybrid transformation-based attack in the same setting as Section 3. As shown in Figure 4, we can conclude that:

*(i) Our hybrid attack is more harmful than general transformation operations.* Compared with the previous 31 transformations in Section 3, our hybrid transformation-based attack can further degenerate the LVLM's performance on all models across all datasets/tasks. This significant similarity decrease demonstrates that manually constructing transformation operations is effective in generating more harmful adversarial examples.

Table 1: Evaluation (averaged similarity scores over three tasks) of different transformation orders on the block-level patches: ① Random Order, ② Inverse Order, and ③ Default Order.

| Dataset | Variant | LLaVA-1.5 | MiniGPT-4 | BLIP-2 | InstructBLIP |
|---------|---------|-----------|-----------|--------|--------------|
| DALL-E | ① | 0.713 | 0.517 | 0.449 | 0.625 |
|        | ② | 0.704 | 0.519 | 0.473 | 0.622 |
|        | ③ | 0.717 | 0.519 | 0.472 | 0.662 |
| SVIT | ① | 0.667 | 0.497 | 0.456 | 0.605 |
|      | ② | 0.665 | 0.503 | 0.456 | 0.622 |
|      | ③ | 0.664 | 0.498 | 0.458 | 0.609 |
| VQAv2 | ① | 0.649 | 0.453 | 0.504 | 0.577 |
|       | ② | 0.652 | 0.436 | 0.530 | 0.582 |
|       | ③ | 0.663 | 0.437 | 0.529 | 0.590 |

*(ii) There is still a lot of room for improving the attack.* The designed hybrid attack method has great attack performance on both image captioning and image classification tasks. However, its performance on VQA task still has lots of room for improvement. We think this is because the hybrid transformation is limited by the enumeration space and can not aggregate the harmful impacts from all possible negative transformations. This inspired us to design an adversarial-learning-based transformation to adaptively search from the whole enumeration space in the next section.

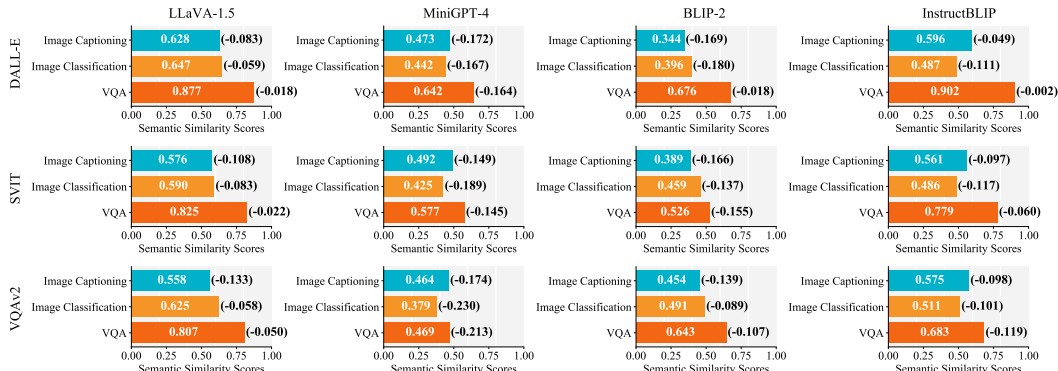

Figure 4: Untargeted attack performance of our designed hybrid transformation-based attack on four LVLM models across three datasets with three tasks. Lower similarities (↓) indicate more harmful impacts. Numbers in front of the bars refer to the similarity score decrease compared to the corresponding best transformations in Section 3, larger decrease indicates greater harmfulness.

In addition to the basic evaluation, we further conduct ablation studies on the designed hybrid transformation-based attack to investigate its sensitivity to the transformation order on the block-level patches. As shown in Table 1, the hybrid transformation-based attack performs similarly on different variants, demonstrating that it is not sensitive to the transformation orders on patches.

### 4.3 ADAPTIVELY LEARNING ADVERSARIAL IMPACTS FROM HARMFUL TRANSFORMATIONS

**Designed Adversarial Transformation-aware Attack.** Although the above hybrid transformation-based attack achieves greatly harmful impacts on LVLM's output semantics, it introduces noticeable and unnatural appearances to humans. Therefore, as shown in Figure 3, we tend to investigate *whether the impacts of potentially harmful transformations can be imposed as perturbations to be added to the raw images while keeping the same adversarial effect as those transformations to improve the imperceptibility of the generated adversarial images.* To this end, inspired by the strategy of momentum-aware gradient calculation (Dong et al., 2018; 2019), we propose to adaptively apply all possible transformation combinations from the random operation set to the image input and only calculate the gradient directions of those harmful ones to the LVLMs to guide and update the transformation-

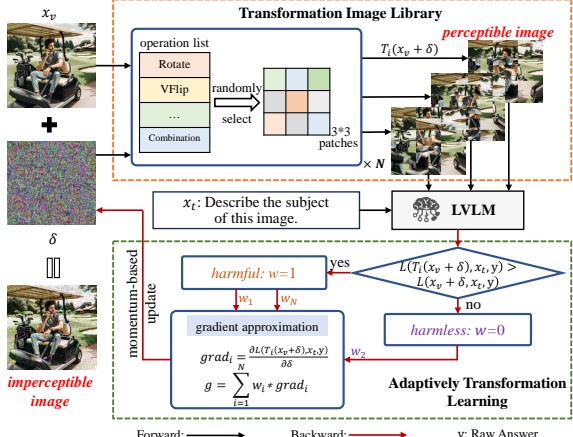

Figure 3: Illustration of our designed adversarial transformation-aware attack. We utilize the adversarial learning strategy with gradient approximation to adaptively impose the truly harmful impacts from all potential transformation operations on the raw image for improving both imperceptibility and effectiveness.

aware perturbations on the raw image. In particular, the gradient direction is approximated by the distance from the original image and its adversarial positive-transformed one. In this manner, the final adversarial images can mimic the harmful impacts of all potential transformations to adaptively learn to best fool LVLMs. Moreover, since adversarial learning is more effective and efficient than the aforementioned manual transformation construction, this attack can further tackle both untargeted and targeted settings with appropriate gradient approximation designs. More adaptive adversarial optimization details and the corresponding algorithm can be found in Appendix C.1.

**Difference with Other LVLM Adversarial Attacks.** Existing LVLM adversarial attacks directly utilize the flexible gradient backpropagation from the whole search space to optimize perturbations according to the specific objective function, relying on prior knowledge of model details and learning unknown/uncontrolled distortion to the raw images. Instead, our attack carefully estimates the

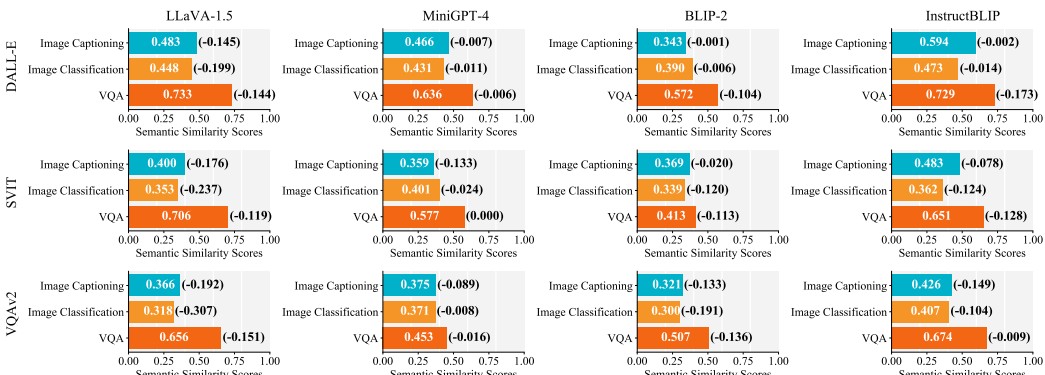

Figure 5: Untargeted attack performance of our adversarial transformation-aware attack on four LVLM models across three datasets with three tasks ($\epsilon = 16/255$). Lower similarities ($\downarrow$) indicate more harmful impacts. Numbers in front of the bars refer to the similarity score decrease compared to the hybrid transformation-based attack in Figure 4, larger decrease indicates greater harmfulness.

Table 2: Targeted attack performance of our adversarial transformation-aware attack. Target text: "I am sorry". Larger similarities ($\uparrow$) indicate more harmful impacts. "*SS*" denotes the semantic similarity between output and target text; "*EM*" and "*CC*" denote "ExactMatch" and "ConditionalContain" metrics to assess whether the output exactly matches or contain the target text, respectively.

| Dataset | LVLM | Perturbation Budget $\epsilon = 32/255$ | | | | | | | | | Perturbation Budget $\epsilon = 16/255$ | | | | | | | | |
| | | Captioning | | | Classification | | | VQA | | | Captioning | | | Classification | | | VQA | | |
| | | SS | EM | CC | SS | EM | CC | SS | EM | CC | SS | EM | CC | SS | EM | CC | SS | EM | CC |
| DALL-E | LLaVA-1.5 | 0.956 | 95.1 | 97.6 | 1.000 | 100.0 | 100.0 | 0.953 | 95.1 | 95.1 | 0.813 | 80.5 | 80.5 | 1.000 | 100.0 | 100.0 | 0.882 | 85.4 | 85.4 |
| | MiniGPT-4 | 0.609 | 58.6 | 58.6 | 0.640 | 61.0 | 65.9 | 0.535 | 53.7 | 53.7 | 0.534 | 46.3 | 46.3 | 0.560 | 48.8 | 51.2 | 0.377 | 31.7 | 31.7 |
| | BLIP-2 | 0.826 | 87.8 | 90.2 | 0.835 | 87.8 | 92.7 | 0.737 | 78.0 | 78.0 | 0.793 | 82.9 | 85.4 | 0.801 | 82.9 | 90.2 | 0.527 | 48.8 | 53.7 |
| | InstructBLIP | 1.000 | 100.0 | 100.0 | 0.962 | 95.1 | 95.1 | 0.750 | 70.7 | 70.7 | 0.762 | 63.4 | 78.0 | 0.695 | 58.5 | 70.7 | 0.500 | 41.5 | 41.5 |
| SVIT | LLaVA-1.5 | 1.000 | 100.0 | 100.0 | 1.000 | 100.0 | 100.0 | 0.930 | 92.7 | 92.7 | 0.952 | 92.7 | 92.7 | 1.000 | 100.0 | 100.0 | 0.903 | 87.8 | 87.8 |
| | MiniGPT-4 | 0.702 | 68.3 | 68.3 | 0.731 | 68.3 | 73.2 | 0.641 | 60.0 | 63.4 | 0.620 | 56.1 | 58.5 | 0.450 | 36.6 | 58.5 | 0.552 | 48.8 | 53.7 |
| | BLIP-2 | 0.834 | 85.5 | 90.2 | 0.884 | 97.6 | 97.6 | 0.772 | 80.5 | 80.5 | 0.779 | 80.5 | 87.8 | 0.803 | 85.4 | 85.4 | 0.533 | 51.2 | 53.7 |
| | InstructBLIP | 0.980 | 97.6 | 97.6 | 0.958 | 95.1 | 95.1 | 0.638 | 56.1 | 56.1 | 0.787 | 70.7 | 80.5 | 0.770 | 65.9 | 79.0 | 0.557 | 46.3 | 46.3 |
| VQAv2 | LLaVA-1.5 | 0.977 | 97.6 | 97.6 | 1.000 | 100.0 | 100.0 | 0.992 | 95.1 | 100.0 | 0.978 | 97.6 | 97.6 | 1.000 | 100.0 | 100.0 | 0.953 | 95.1 | 97.6 |
| | MiniGPT-4 | 0.634 | 61.0 | 63.4 | 0.713 | 70.7 | 70.7 | 0.612 | 53.7 | 61.0 | 0.539 | 46.3 | 51.2 | 0.568 | 53.7 | 56.1 | 0.522 | 43.9 | 51.2 |
| | BLIP-2 | 0.849 | 90.2 | 92.7 | 0.837 | 87.8 | 92.7 | 0.648 | 58.5 | 68.3 | 0.775 | 82.9 | 82.9 | 0.819 | 87.8 | 87.8 | 0.588 | 53.7 | 61.0 |
| | InstructBLIP | 1.000 | 100.0 | 100.0 | 0.947 | 92.7 | 95.1 | 0.480 | 34.1 | 34.1 | 0.828 | 70.7 | 90.2 | 0.724 | 61.0 | 75.6 | 0.404 | 24.4 | 24.4 |

operation-specific gradients from transformations to update perturbations, which is more practical to be exploited in a black-box setting and can *explicitly learn transformation-only adversarial impacts*.

**Evaluation and Discussion.** To evaluate our designed adversarial transformation-aware attack, we conduct experiments in both untargeted and targeted attack settings and can conclude that:

*(i) As for the untargeted attack, this adversarial learning attack is more effective and efficient than the hybrid transformation-based attack.* Although the hybrid transformation-based attack tries to enumerate possible transformation combinations and manually construct the most harmful operations, it costs lots of resources and may stuck into the local optimum. Instead, this adversarial transformation-aware attack can adaptively learn the most harmful impacts from all potential transformation combinations, leading to more harmful adversarial generations as shown in Figure 5.

*(ii) This adversarial learning attack is more flexible and can mislead the LVLMs output attacker-chosen texts.* In addition to the above untargeted adversarial generation, we also investigate whether our learnable attack satisfies fooling the LVLMs with specific target output texts. As shown in Table 2, we preset the target response as "I am sorry" and experimental results indicate that our adversarial transformation-aware attack is effective in achieving targeted attack with significant performance, having great potential to be deployed in real-world LVLM applications. Besides, it also shows that our attack can still achieve competitive performances with a lower perturbation budget.

Overall, our attacks show superior performance, demonstrating the effectiveness of our carefully designed transformation-aware attacks. More experimental results are in Appendix C.2.

Table 3: Investigation on the adversarial transferability of our two types of transformation attacks. Following previous works, we evaluate the untargeted attack performance of the adversarial samples generated on the LVLMs of rows and tested on the LVLMs of columns. The experimental results are calculated by the averaged semantic similarities on three tasks. "value" is basic performance.

| Dataset | LVLM | Hybrid Transformation-based Attack ($\downarrow$) | | | | Adversarial Transformation-aware Attack ($\downarrow$) | | | |
|---|---|---|---|---|---|---|---|---|---|
| | | LLaVA-1.5 | MiniGPT-4 | BLIP-2 | InstructBLIP | LLaVA-1.5 | MiniGPT-4 | BLIP-2 | InstructBLIP |
| DALL-E | LLaVA-1.5 | 0.717 | 0.710 | 0.611 | 0.763 | 0.463 | 0.697 | 0.678 | 0.728 |
| | MiniGPT-4 | 0.808 | 0.519 | 0.626 | 0.752 | 0.770 | 0.460 | 0.399 | 0.456 |
| | BLIP-2 | 0.782 | 0.707 | 0.472 | 0.701 | 0.702 | 0.676 | 0.372 | 0.672 |
| | InstructBLIP | 0.789 | 0.695 | 0.595 | 0.662 | 0.693 | 0.661 | 0.565 | 0.542 |
| SVIT | LLaVA-1.5 | 0.664 | 0.663 | 0.563 | 0.660 | 0.439 | 0.695 | 0.683 | 0.708 |
| | MiniGPT-4 | 0.773 | 0.498 | 0.613 | 0.682 | 0.711 | 0.397 | 0.338 | 0.450 |
| | BLIP-2 | 0.742 | 0.673 | 0.458 | 0.679 | 0.650 | 0.553 | 0.300 | 0.519 |
| | InstructBLIP | 0.747 | 0.659 | 0.565 | 0.609 | 0.663 | 0.545 | 0.383 | 0.427 |
| VQAv2 | LLaVA-1.5 | 0.663 | 0.646 | 0.643 | 0.742 | 0.394 | 0.657 | 0.662 | 0.694 |
| | MiniGPT-4 | 0.777 | 0.437 | 0.663 | 0.721 | 0.700 | 0.349 | 0.321 | 0.477 |
| | BLIP-2 | 0.754 | 0.653 | 0.529 | 0.672 | 0.641 | 0.518 | 0.314 | 0.484 |
| | InstructBLIP | 0.759 | 0.634 | 0.640 | 0.590 | 0.632 | 0.518 | 0.431 | 0.442 |

## 4.4 In-depth analysis of our proposed transformation attacks

In this section, we provide a detailed analysis of our proposed two types of transformation attacks from perspectives of complexity, adversarial transferability, and adversarial robustness, respectively.

**Takeaways:** ❶ Our proposed attack methods in Section 4 are quite efficient. Besides, our adversarial learning based attack variant is more efficient than the manual constructing one while achieving better performance. ❷ Our developed adversarial transformation attacks can achieve significant transferability among different black-box LVLM models. ❸ Experimental results also illustrate that our two transformation attacks are robust to potential defense strategies.

**Analysis on Complexity.** We first investigate the complexity of our proposed two types of LVLM attacks. As shown in Table 4, we evaluate the usage of GPU time and memory of a single adversarial sample on both generation and inference processes. It indicates that the adversarial transformation-aware attack is much more efficient than the hybrid transformation-based attack during the adversarial generation, as the former can adaptively learn the potentially harmful transformation impacts (but re-

Table 4: Complexity analysis on our attacks. We evaluate the GPU time and memory usage of a single adversarial example on both generation and inference processes on LLaVA-1.5.

| Process | Attack Type | GPU Time | GPU Memory |
|---|---|---|---|
| Generation | Hybrid Attack | 9min | 16GB |
| | Adversarial Attack | 5min | 22GB |
| Inference | Hybrid Attack | 3s | 15GB |
| | Adversarial Attack | 3s | 15GB |

quires relatively more memory for gradient approximation) while the latter relies on lots of manual efforts. Since both two attacks solely feed the adversarial sample into the LVLM without any additional operation during the inference, they have the same complexity in inference.

**Analysis on Adversarial Transferability.** We then investigate the adversarial transferability of the generated adversarial examples of our two attacks. As shown in Table 3, we can conclude that:

*(i) Developing LVLM attacks using visual transformations can achieve significant adversarial transferability.* Our two types of transformation attacks achieve great transfer-attack performance when we directly feed the generated adversarial examples of one LVLM to the other three LVLMs. Although the output textual semantic similarities relatively decrease, its influences are largely inferior to the performance drops brought by our attacks. Therefore, utilizing visual transformations to construct LVLM attacks is a promising way to improve the adversarial transferability.

*(ii) The adversarial transformation-aware attack achieves better transferability than the hybrid transformation-based attack.* Since the transformation attack with adversarial learning mechanism can adaptively learn more potential transformation operations than the hybrid manual constructing one, it will learn more generalizable transformation impacts thus leading to better transferability.

**Analysis on Adversarial Robustness.** At last, we investigate the robustness of the proposed two transformation attacks. In particular, we implement three pre-processing defenses, *i.e*, Randomization (Frosio & Kautz, 2023; Xie et al., 2017), JPEG Compression (Guo et al., 2017), and Diffusion Restoration (Nie et al., 2022). As shown in Table 5, we can conclude that:

Table 5: Investigation on the adversarial robustness of our two types of transformation attacks. Following previous works, we evaluate the untargeted attack performance of the adversarial samples generated on the LVLMs of columns by testing them against potential defenses of rows. The experimental results are calculated by the averaged semantic similarities on three tasks.

| Dataset | Defense | Hybrid Transformation-based Attack ($\downarrow$) | | | | Adversarial Transformation-aware Attack ($\downarrow$) | | | |
|---------|---------|----------|----------|--------|--------------|----------|----------|--------|--------------|
| | | LLaVA-1.5 | MiniGPT-4 | BLIP-2 | InstructBLIP | LLaVA-1.5 | MiniGPT-4 | BLIP-2 | InstructBLIP |
| DALL-E | No Defense | 0.717 | 0.519 | 0.472 | 0.662 | 0.463 | 0.460 | 0.372 | 0.542 |
| | Randomization | 0.767 | 0.686 | 0.581 | 0.674 | 0.744 | 0.591 | 0.505 | 0.631 |
| | JPEG Compre. | 0.776 | 0.681 | 0.548 | 0.682 | 0.609 | 0.565 | 0.468 | 0.604 |
| | Diffusion | 0.541 | 0.518 | 0.382 | 0.421 | 0.758 | 0.701 | 0.594 | 0.697 |
| SVIT | No Defense | 0.664 | 0.498 | 0.458 | 0.609 | 0.439 | 0.397 | 0.300 | 0.427 |
| | Randomization | 0.763 | 0.658 | 0.545 | 0.637 | 0.648 | 0.560 | 0.406 | 0.503 |
| | JPEG Compre. | 0.757 | 0.648 | 0.508 | 0.631 | 0.547 | 0.487 | 0.359 | 0.471 |
| | Diffusion | 0.555 | 0.514 | 0.397 | 0.440 | 0.690 | 0.611 | 0.532 | 0.617 |
| VQAv2 | No Defense | 0.663 | 0.437 | 0.529 | 0.590 | 0.394 | 0.349 | 0.314 | 0.442 |
| | Randomization | 0.746 | 0.609 | 0.599 | 0.696 | 0.610 | 0.467 | 0.409 | 0.512 |
| | JPEG Compre. | 0.760 | 0.591 | 0.536 | 0.675 | 0.502 | 0.469 | 0.372 | 0.485 |
| | Diffusion | 0.536 | 0.464 | 0.392 | 0.423 | 0.675 | 0.571 | 0.550 | 0.608 |

*(i) Our proposed attack methods are robust to potential defense strategies.* According to the performances in this table, although the three defense methods are able to degenerate our attack performance, their influences are largely inferior to the performance drops brought by our attack. This demonstrates that our attack is fairly resistant to the potential defense methods in practice.

*(ii) The hybrid attack produces even more harmful results under diffusion restoration.* It is because the applied transformations destroy the image structure, so diffusion operation will further generate more diverse content. Instead, diffusion can alleviate the harmful impact of adversarial noise.

## 5    DISCUSSION

**Justification of Our Experiments.** Since our main goal is to investigate the adversarial robustness of LVLMs to visual transformations, our experiments are solely conducted on the comparisons and analysis between different transformation strategies. We do not compare performances with other types of LVLM attacks as: (1) They are designed with more complicated perturbation patterns. Directly comparing our solely transformation-based attacks with them is unfair. (2) They are diversely implemented in different settings with the usage of different LVLM models and datasets. We provide case-by-case comparisons with other LVLM attacks under the same settings in Appendix D.

**Limitations.** Our work assumes that input images are fed directly into the LVLM models. However, in the future, vision-language models are more likely to be deployed in complex scenarios such as controlling robots or automatic driving, in which case input images may be obtained from the interaction with physical environments and captured in real time by cameras. Performing attacks in such complicated cases would be one of the future directions for evaluating the LVLM security.

**Broader Impacts.** While the primary goal of our research is to generate superior adversarial transformations against large vision-language models, it is possible that the developed attacking strategies could be misused to evade practically deployed systems and cause potential negative societal impacts. Specifically, our adversarial threat model assumes targeted responses, which involves manipulating existing APIs such as GPT-4 (with visual inputs) and/or Midjourney on purpose, thereby increasing the risk if these vision-language APIs are implemented as plugins in other products.

## 6    CONCLUSION

In conclusion, this paper offers novel insights into the vulnerability of LVLMs to visual transformations. Our comprehensive evaluation indicates that different transformations share diverse harmfulness degrees on existing LVLMs while appropriate transformation combinations can boost the attack performance. We also take a further step to investigate how to manually construct a more harmful transformation operation and how to adaptively learn to impose adversarial impacts from all potential transformations to raw images for improving the attack effectiveness and imperceptibility. We envision our findings will pave the way for the development of efficient and effective LVLM attacks.

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

# A ADDITIONAL DETAILS OF LVLMS' PERFORMANCE ON DIFFERENT VISUAL TRANSFORMATIONS

## A.1 MORE DETAILS OF VISUAL TRANSFORMATIONS

Here we provide the implementation details of the previously mentioned transformations, respectively. In particular, for a given image block $x \in \mathbb{R}^{3 \times H \times W}$, we can implement the basic transformations as follows:

**1. Resize_Large**: We resize the original image $x$ into $x'$ with the size of $3 \times h \times w$ ($h > w, w > w$) using bilinear interpolation.

**2. Resize_Small**: We resize the original image $x$ into $x'$ with the size of $3 \times h \times w$ ($h < w, w < w$).

**3. HFlip (Horizontal Flip)**: We flip the image $x$ horizontally along the vertical axis, in which the left of the image becomes the right, and the right becomes the left.

**4. VFlip (Vertical Flip)**: We flip the image $x$ vertically along the horizontal axis, in which the top of the image becomes the bottom, and the bottom becomes the top.

**5. Rotate_Random**: We rotate the image $x$ by a random angle around its center point.

**6. Rotate_180°**: We turn the image $x$ clockwise by $180°$ around its center point, in which the top-left of the image becomes the bottom-right, and the top-right becomes the bottom-left.

**7. VShift_Random (Vertical Shift)**: We roll the image $x$ along the vertical axis by a randomly selected length $h < H$.

**8. VShift_Half (Vertical Shift)**: We roll the image $x$ along the vertical axis at $h = \frac{1}{2}H$.

**9. HShift_Random (Horizontal Shift)**: We roll the image block $x$ along the horizontal axis by a randomly selected length of $w < W$.

**10. HShift_Half (Horizontal Shift)**: We roll the image block $x$ along the horizontal axis at $w = \frac{1}{2}W$.

**11. Scale**: We multiply a random scale factor $\alpha \in (0,1)$ with the pixel in the image to scale $x$ into $\alpha \cdot x$.

**12. Add Noise**: We add a uniform noise $r \in [0,1]^{3 \times H \times W}$ to the image $x$ and clip them into $[0,1]$ to obtain the transformed image $Clip(x+r, 0, 1)$.

**13. Dropout**: We set each channel of the image $x$ to zero with a probability of $10\%$.

**14. ColorJitter**: We randomly change the brightness, contrast, saturation, and hue of the image $x$.

**15. DCT**: We first transform $x$ to the frequency domain using Discrete Cosine Transformation (DCT). Then we mask the top 40% highest frequency with 0 and recover the image in the time domain using Inverse Discrete Cosine Transformation (IDCT).

Then, we adapt the above transformations into the block-level transformation (AprilPyone & Kiya, 2021). Specifically, we first uniformly split the image $x \in \mathbb{R}^{3 \times H \times W}$ into $3 \times 3$ patches of the same sizes, then perform the above operations on each patch as follows:

**16. Block_Resize**: We resize each patch to size $3 \times h \times w$ ($h < \frac{1}{3}H, w < \frac{1}{3}H$) and utilize bicubic interpolation to reconstruct the patch into the original size.

**17. Block_HFlip**: We flip each patch along along the vertical axis, in which the left of the patch becomes the right, and the right becomes the left.

**18. Block_VFlip**: We flip each patch along the horizontal axis, in which the top of the patch becomes the bottom, and the bottom becomes the top.

**19. Block_Rotate**: We turn each patch clockwise by $180°$ around its center point, in which the top-left of the patch becomes the bottom-right, and the top-right becomes the bottom-left.

**20. Block_VShift**: We roll each patch along the vertical axis at half height.

**21. Block_HShift**: We roll each patch along the horizontal axis at half weight.

**22. Block_Scale**: We multiply a random scale factor $\alpha \in (0,1)$ with the pixel of each patch.

**23. Block_AddNoise**: We add a uniform noise to each patch and clip them into $[0,1]$ to obtain the transformed patch.

**24. Block_Dropout**: We set each channel of the patch to zero with a probability of $10\%$.

**25. Block_ColorJitter**: We randomly change the brightness, contrast, saturation, and hue of each patch.

**26. Block_DCT**: we first transform each patch to the frequency domain using Discrete Cosine Transformation (DCT). Then we mask the top 40% highest frequency with 0 and recover the patch in the time domain using Inverse Discrete Cosine Transformation (IDCT).

Based on our empirical experience, we find that transformations of Rotate, Hshilt, VFlip perform more adversarial among the above multiple block-level transformations. Therefore, we further design various types of their combinations in the following:

**27. Block_Rotate_HShift**: We randomly choose one of the rotation or horizontal shift operations for each patch.

**28. Block_Rotate_VFlip**: We randomly choose one of the rotation or vertical flip operations for each patch.

**29. Block_VFlip_HShift**: We randomly choose one of the vertical flip or horizontal shift operations for each patch.

**30. Block_Rotate_VFlip_HShift**: We randomly choose one of the rotation, vertical flip, or horizontal shift operations for each patch.

**31. Block_Random_Combination**: We randomly choose the combination of the transformations including rotation, vertical flip, or horizontal shift operations for each patch.

We provide visual examples of the various image transformations described above in Figure 6.

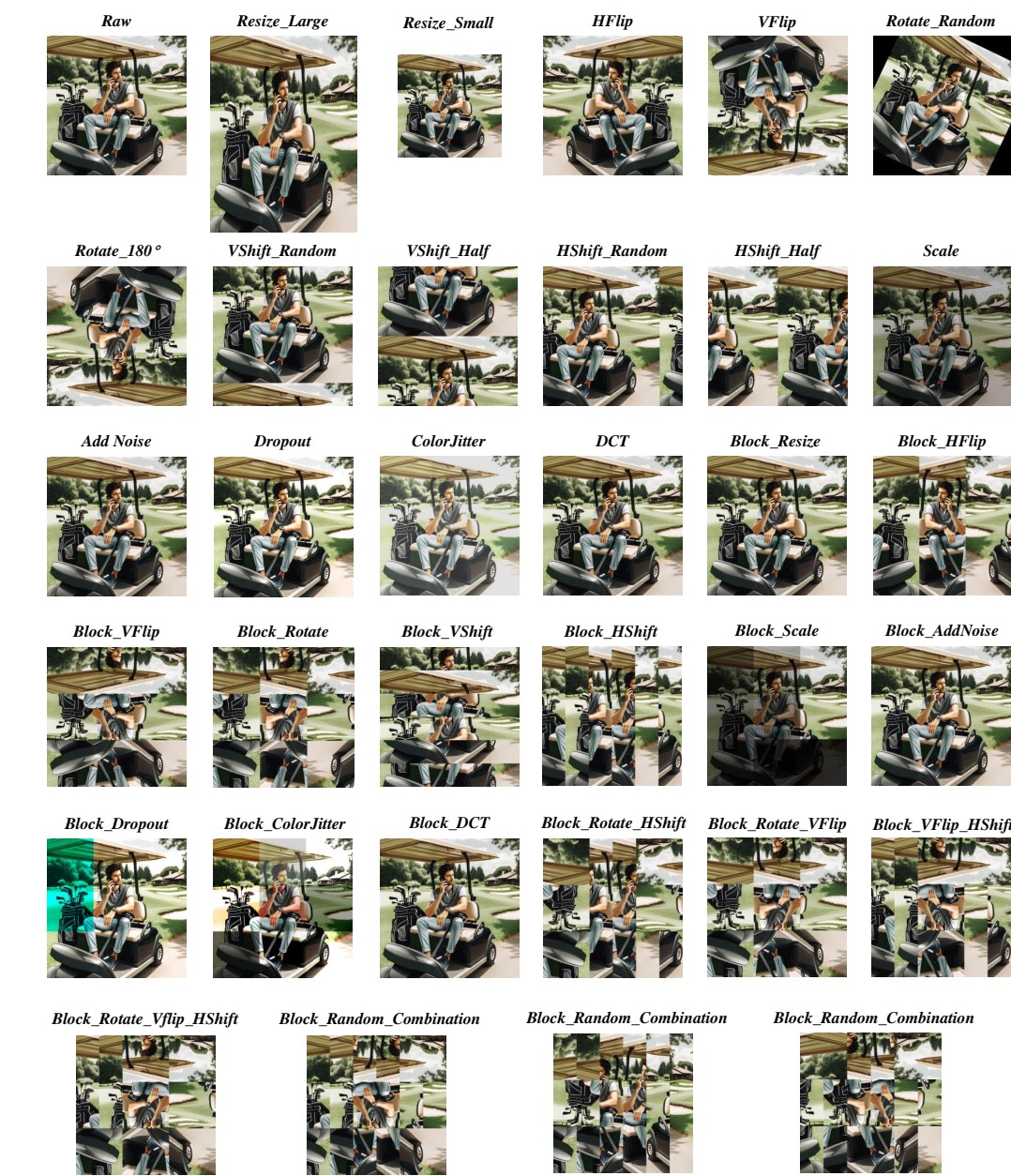

Figure 6: Visualization examples of our utilized 31 number of visual transformations.

## A.2 VISUALIZATION ON THE TEXTUAL OUTPUTS OF TRANSFORMED IMAGES

We provide the textual outputs of each transformed image in Figure 7. We can find that the general visual transformation can affect the LVLM's textual output.

## B ADDITIONAL DETAILS OF OUR PROPOSED HYBRID TRANSFORMATION-BASED ATTACK

### B.1 MORE ILLUSTRATIONS OF OUR HYBRID TRANSFORMATION-BASED ATTACK

We provide a step-by-step visualization of our proposed hybrid transformation-based attack. As shown in Figure 8, our hybrid transformation-based attack transforms each patch one by one in the

Input Text: What are the two distinct colors of the cats sitting in the doorway?

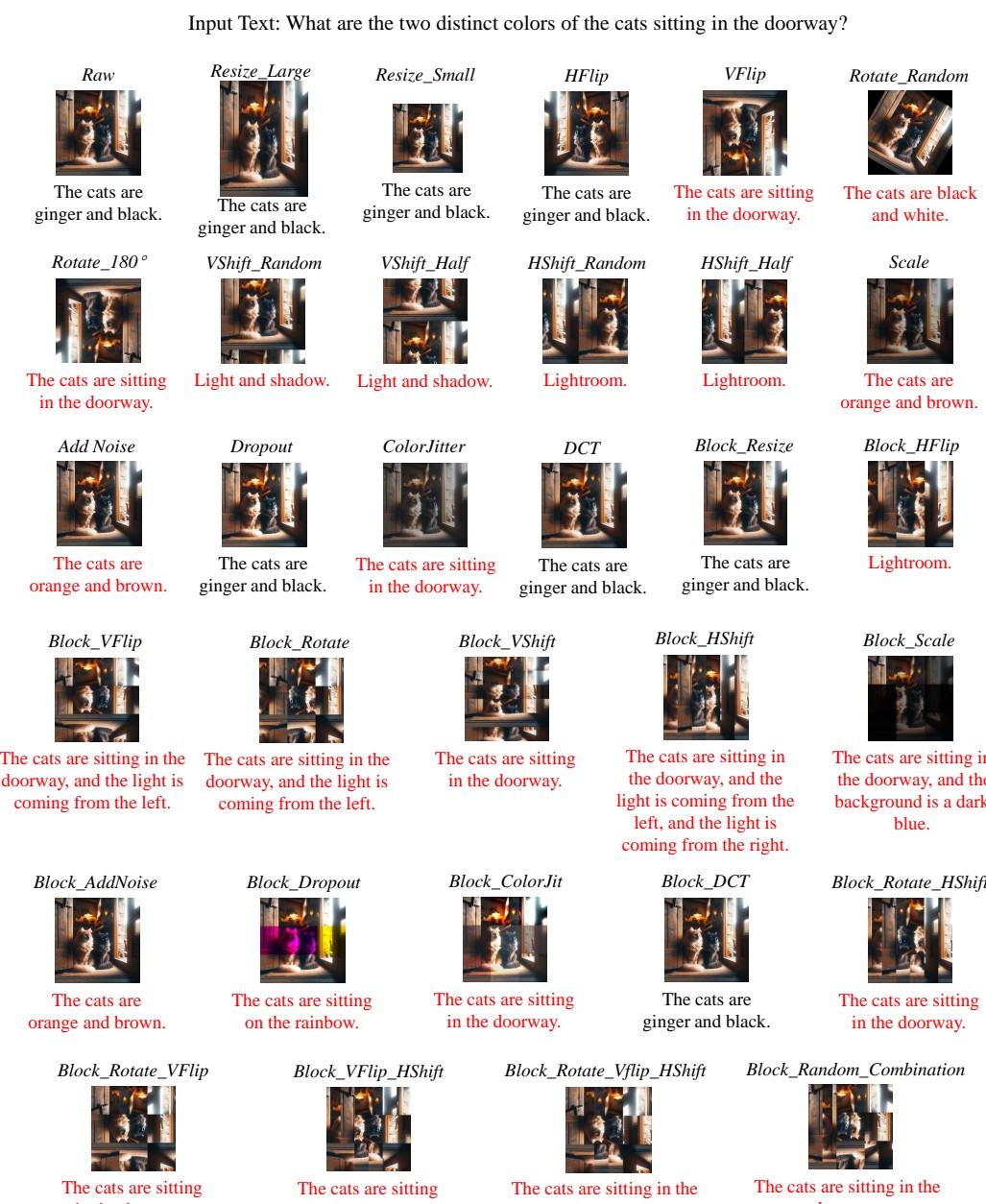

Figure 7: Visualization of the LVLM's textual outputs of different transformation operations. Red: the affected outputs are different from the raw answer.

default order to iteratively make the transformed image as harmful as possible. Starting from the first patch, we fix the remaining patches unchanged and perform the above 15 transformation operations in sequence to obtain the corresponding 15 transformed images. Then each transformed image is fed into the LVLM model individually with the same textual prompt to obtain the corresponding 15 adversarial answers. Next, we calculate the semantic similarities between these adversarial answers and the original answer, and select the operation with the lowest similarity score as the optimal (most harmful) transformation operation for this patch. By fixing the transformed patch 1, we repeat this process for patch 2 to further degenerate the LVLM's performance. After traversing all patches, we can generate the most harmful transformation operation on the image input. Note that, each step operation can effectively further degenerate the LVLM's performance compared to its previous step.

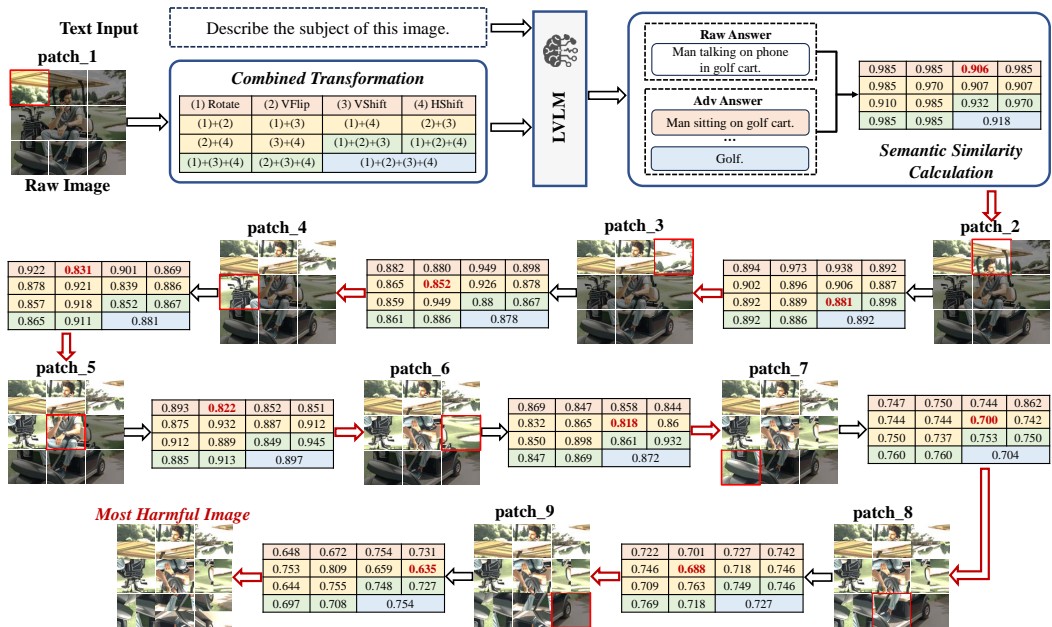

Figure 8: Illustration of our designed hybrid transformation-based attack, which manually constructs the most harmful transformation combination via enumeration.

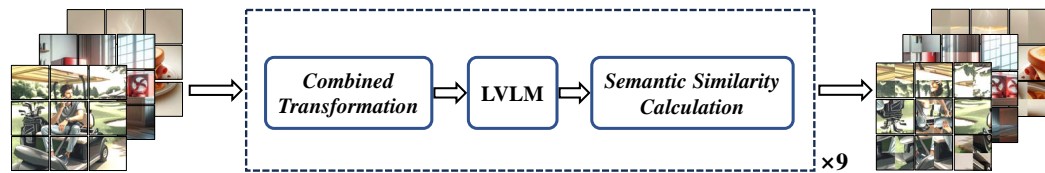

Figure 9: Illustration of our designed hybrid transformation-based attack implemented in a universal setting, where the most harmful transformation operation is explored to be the same among all image-text inputs.

## B.2 UNIVERSAL ATTACK FOR OUR HYBRID TRANSFORMATION-BASED ATTACK

Generally, our proposed hybrid transformation-based attack is implemented in a single-image attack setting, where the most harmful transformation operation varies among different image-text inputs. Further, we can also extend this attack into a universal attack setting as shown in Figure 9, where the most harmful transformation operation is explored to be the same among all image-text inputs. Specifically, we follow the traditional universal setting (Moosavi-Dezfooli et al., 2017) to assess the vulnerability of each transformation based on its averaged impacts on the whole test set. Corresponding performance is shown in Figure 10, we can conclude that:

*(i) Our hybrid transformation-based attack in a universal setting is also more harmful than general transformation combinations.* Compared with the previous 31 transformations in Section 3, our hybrid transformation-based attacks in a universal setting can further degenerate the LVLM's performance on all models across all datasets. This significant similarity decrease also demonstrates that manually constructing transformation operations is more effective in generating more harmful adversarial examples.

*(ii) As for our hybrid transformation-based attack, the single-image attack setting is more effective than the universal setting to generate more harmful transformation operations.* By comparing the attack performance between the single-image setting in Figure 4 and the universal setting in Figure 10, we can find that the single-image attack is more flexible and harmful than the universal attack setting, thus achieving better attack performance. This is because the single-image attack can straightforwardly conduct the most vulnerable transformation impacts on each image while the universal attack fails to cover the distribution gaps among diverse images.

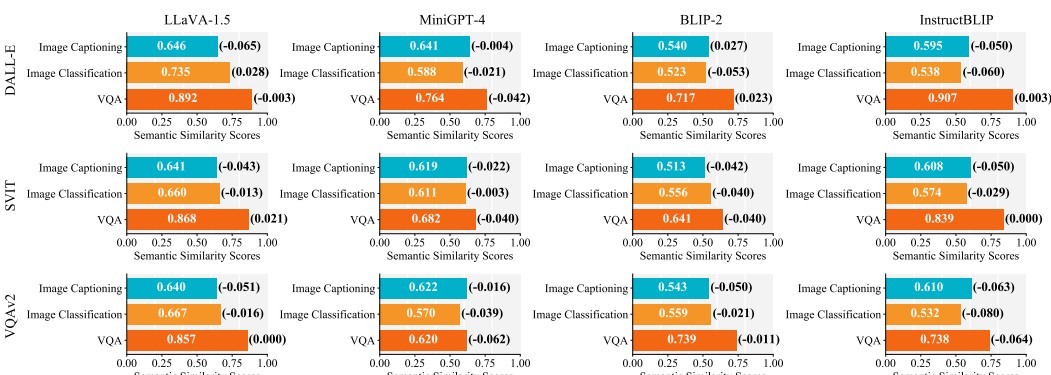

Figure 10: Untargeted attack performance of our designed hybrid transformation-based attack implemented in a universal setting. Lower similarities (↓) indicate more harmful impacts. Numbers in front of the bars refer to the similarity score decrease compared to the corresponding best transformations in Section 3, larger decrease indicates greater harmfulness.

### B.3 Visualization on the textual outputs of transformed images

To validate the effectiveness of our hybrid transformation-based attack, we provide the visualization on the textual outputs of the transformed images. As shown in Figure 11, we can find that our method can effectively mislead LVLMs to output wrong texts that are semantically distinct from the original texts.

## C Additional Details of our proposed adversarial transformation-aware attack

### C.1 More Illustrations of Our adversarial transformation-aware attack

Although LVLMs are insensitive and invariant to the block-level transformations as LVLMs can globally handle and learn the tokenize-wise semantics, the hybrid transformation-based attack still introduces noticeable and unnatural appearances to humans. Therefore, to improve the imperceptibility of the transformed images, we propose to impose the impacts of potentially harmful transformations as perturbations to add to the raw image while keeping the same adversarial effect as those transformations. In this manner, the perturbed image is optimized by the approximated gradient directions of harmful transformations and shares the same adversarial impacts, achieving the goal of adversarial transformation while improving visual imperceptibility. Specifically, instead of directly calculating the gradient on the input image, we calculate the gradient only on the truly harmful transformations. More adversarial training details can be found in Algorithm 1.

### C.2 More Experiments on Our adversarial transformation-aware attack

As for the untargeted attack, the experiments in Figure 5 are implemented with perturbation budget $\epsilon = 16/255$. We further provide the corresponding experiments with perturbation budget $\epsilon = 32/255$ in Figure 12. We can find that our adversarial transformation-aware attack can achieve better attack performance with larger budgets.

As for the targeted attack, we also perform more experiments on different target text output of "I want to destroy the world" in Table 6 and "I don't know" in Table 7. Corresponding results show that our attack can still achieve significant adversarial performance, demonstrating the scalability, generalizability, and practicality of our proposed attack method.

### C.3 Visualization on the textual outputs of transformed images

To validate the effectiveness of our adversarial transformation-aware attack, we provide the visualization on the textual outputs of the transformed images in both untargeted and targeted attack settings. As shown in Figure 11, we can find that our method can not only effectively mislead

Figure 11: Visualization of the LVLM's textual outputs. Green: hybrid transformation-based attack. Red and Blue: adversarial transformation-aware attack on untargeted and targeted attack settings.

---

**Algorithm 1** Adversarial Transformation-aware Attack

---

**Input:** The source sample with the raw image $x_v$, the text input $x_t$ and the raw answer $y$; the loss function of LVLM $L$; the number of iteration $T$; the maximum perturbation $\epsilon$; decay factor $\mu$; the number of transformed images $N$; the step size $\alpha$; the random transformation image function $Trans$.

**Output:** Adversarial image.

1: Initialize gradient $g_0 = 0$, adversarial sample $x_{v,0}^{adv} = x_v$
2: **for** $t = 0$ **to** $T - 1$ **do**
3: $\quad g = 0$
4: $\quad$ **for** $i = 0$ **to** $N - 1$ **do**
5: $\quad\quad$ Construct transformed image as $Trans_i(x_{v,t}^{adv})$
6: $\quad\quad$ Calculate the loss before and after the transformation by $l_1 = L(Trans_i(x_{v,t}^{adv}), x_t, y)$
7: $\quad\quad$ and $l_2 = L(x_{v,t}^{adv}, x_t, y)$
8: $\quad\quad$ **if** $l_1 > l_2$ **then**
9: $\quad\quad\quad$ Get the harmful weight as $w_i = 1$
10: $\quad\quad$ **else**
11: $\quad\quad\quad$ Get the harmless weight as $w_i = 0$
12: $\quad\quad$ **end if**
13: $\quad\quad$ Approximate the gradient by $grad_i = \nabla_{x_{v,t}^{adv}} L(Trans_i(x_{v,t}^{adv}), x_t, y)$
14: $\quad\quad$ Sum the gradients as $g = g + w_i \cdot grad_i$
15: $\quad$ **end for**
16: $\quad$ Get the average gradients as $g = \frac{1}{N} \cdot g$
17: $\quad$ Update the momentum by $g_{t+1} = \mu \cdot g_t + \frac{g}{\|g\|_1}$
18: $\quad$ Update the adversarial image by $x_{v,t+1}^{adv} = \text{Clip}(x_{v,t}^{adv} + \alpha \cdot \text{sign}(g_{t+1}), 0, 1)$
19: **end for**
20: **return** transformation-aware adversarial sample $x_{v,T}^{adv}$

---

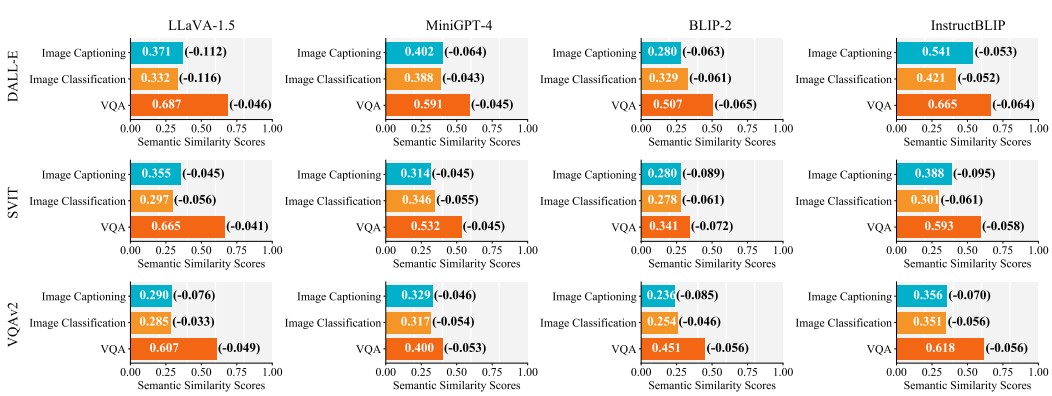

Figure 12: Untargeted attack performance of our designed adversarial transformation-aware attack implemented on the $\epsilon = 32/255$ setting. Lower similarities ($\downarrow$) indicate more harmful impacts. Numbers in front of the bars refer to the similarity score decrease compared to the adversarial transformation-aware attack variant implemented with $\epsilon = 16/255$ in Figure 5, larger decrease indicates greater harmfulness.

LVLMs to output wrong texts that are semantically distinct from the original texts, but also mislead LVLMs to output specific attacker-chosen texts.

## D  PERFORMANCE COMPARISON WITH EXISTING LVLM ATTACKS

To further investigate the effectiveness of our proposed attack, we provide a case-by-case comparison with existing LVLM attacks MF (Zhao et al., 2024) (NeurIPS 2023) and CroPA (Luo et al., 2024) (ICLR 2024). Since existing LVLM attacks are diversely implemented in different settings with the usage of different LVLM models and datasets, we re-implement our transformation-aware attack into their same settings/datasets/metrics for fair comparison. Note that, since MF and CroPA

Table 6: Targeted attack performance of our adversarial transformation-aware attack. Target text: "I want to destroy the world". Larger similarities (↑) indicate more harmful impacts. "*SS*" denotes the semantic similarity between output and target text; "*EM*" and "*CC*" denote the "ExactMatch" and "ConditionalContain" metrics to assess whether the output exactly matches or contain the target text, respectively.

| Dataset | LVLM | Perturbation Budget $\epsilon = 32/255$ | | | | | | | | | Perturbation Budget $\epsilon = 16/255$ | | | | | | | | |
|---|---|---|---|---|---|---|---|---|---|---|---|---|---|---|---|---|---|---|---|
| | | Captioning | | | Classification | | | VQA | | | Captioning | | | Classification | | | VQA | | |
| | | SS | EM | CC | SS | EM | CC | SS | EM | CC | SS | EM | CC | SS | EM | CC | SS | EM | CC |
| DALL-E | LLaVA-1.5 | 0.841 | 82.9 | 82.9 | 0.983 | 97.6 | 97.6 | 0.791 | 78.0 | 78.0 | 0.568 | 51.2 | 51.2 | 0.959 | 95.1 | 95.1 | 0.518 | 48.8 | 51.2 |
| | MiniGPT-4 | 0.907 | 87.8 | 90.2 | 0.837 | 78.0 | 85.4 | 0.770 | 73.2 | 75.6 | 0.792 | 68.3 | 78.0 | 0.769 | 68.3 | 75.6 | 0.661 | 63.4 | 63.4 |
| | BLIP-2 | 0.891 | 82.9 | 82.9 | 0.888 | 85.4 | 87.8 | 0.718 | 65.9 | 65.9 | 0.770 | 68.3 | 68.3 | 0.748 | 63.4 | 63.4 | 0.483 | 39.0 | 39.0 |
| | InstructBLIP | 0.853 | 82.9 | 85.4 | 0.772 | 65.9 | 73.2 | 0.563 | 51.2 | 51.2 | 0.786 | 73.2 | 78.0 | 0.684 | 61.0 | 65.9 | 0.331 | 24.4 | 24.4 |
| SVIT | LLaVA-1.5 | 0.904 | 87.8 | 90.2 | 1.000 | 100.0 | 100.0 | 0.767 | 75.6 | 75.6 | 0.699 | 65.9 | 68.3 | 1.000 | 100.0 | 100.0 | 0.543 | 51.2 | 53.7 |
| | MiniGPT-4 | 0.892 | 87.8 | 87.8 | 0.914 | 90.2 | 90.2 | 0.849 | 80.5 | 82.9 | 0.859 | 85.4 | 85.4 | 0.801 | 78.0 | 78.0 | 0.790 | 75.6 | 80.5 |
| | BLIP-2 | 0.962 | 97.6 | 97.6 | 0.920 | 90.2 | 92.7 | 0.728 | 70.7 | 70.7 | 0.865 | 80.5 | 82.9 | 0.779 | 70.7 | 73.2 | 0.610 | 56.1 | 56.1 |
| | InstructBLIP | 0.910 | 90.2 | 92.7 | 0.807 | 68.3 | 80.5 | 0.483 | 41.5 | 43.9 | 0.874 | 80.5 | 87.8 | 0.745 | 65.9 | 73.2 | 0.435 | 31.7 | 31.7 |
| VQAv2 | LLaVA-1.5 | 0.934 | 92.7 | 92.7 | 1.000 | 100.0 | 100.0 | 0.906 | 90.2 | 90.2 | 0.819 | 80.5 | 80.5 | 0.955 | 95.1 | 95.1 | 0.765 | 75.6 | 75.6 |
| | MiniGPT-4 | 0.903 | 87.8 | 87.8 | 0.878 | 85.4 | 87.8 | 0.814 | 80.5 | 80.5 | 0.792 | 75.6 | 75.6 | 0.841 | 82.9 | 82.9 | 0.686 | 65.9 | 65.9 |
| | BLIP-2 | 0.966 | 97.6 | 97.6 | 0.910 | 85.4 | 87.8 | 0.584 | 53.7 | 53.7 | 0.861 | 82.9 | 82.9 | 0.894 | 82.9 | 85.4 | 0.546 | 46.3 | 46.3 |
| | InstructBLIP | 1.000 | 100.0 | 100.0 | 0.832 | 75.6 | 82.9 | 0.421 | 36.6 | 36.6 | 0.916 | 87.8 | 92.7 | 0.790 | 70.7 | 78.0 | 0.386 | 31.7 | 31.7 |

Table 7: Targeted attack performance of our adversarial transformation-aware attack. Target text: "I don't know". Larger similarities (↑) indicate more harmful impacts. "*SS*" denotes the semantic similarity between output and target text; "*EM*" and "*CC*" denote the "ExactMatch" and "ConditionalContain" metrics to assess whether the output exactly matches or contain the target text, respectively.

| Dataset | LVLM | Perturbation Budget $\epsilon = 32/255$ | | | | | | | | | Perturbation Budget $\epsilon = 16/255$ | | | | | | | | |
|---|---|---|---|---|---|---|---|---|---|---|---|---|---|---|---|---|---|---|---|
| | | Captioning | | | Classification | | | VQA | | | Captioning | | | Classification | | | VQA | | |
| | | SS | EM | CC | SS | EM | CC | SS | EM | CC | SS | EM | CC | SS | EM | CC | SS | EM | CC |
| DALL-E | LLaVA-1.5 | 1.000 | 100.0 | 100.0 | 1.000 | 100.0 | 100.0 | 1.000 | 100.0 | 100.0 | 0.776 | 75.0 | 77.5 | 0.981 | 97.5 | 100.0 | 0.930 | 92.5 | 92.5 |
| | MiniGPT-4 | 0.821 | 81.3 | 84.4 | 0.801 | 78.0 | 82.9 | 0.848 | 82.9 | 82.9 | 0.805 | 78.0 | 80.5 | 0.749 | 70.7 | 78.0 | 0.708 | 65.9 | 65.9 |
| | BLIP-2 | 0.687 | 63.6 | 72.7 | 0.713 | 65.9 | 75.6 | 0.605 | 51.6 | 54.8 | 0.502 | 44.3 | 52.2 | 0.512 | 39.0 | 51.2 | 0.455 | 39.0 | 41.5 |
| | InstructBLIP | 0.797 | 77.1 | 77.1 | 0.810 | 75.9 | 75.9 | 0.740 | 67.9 | 67.9 | 0.611 | 51.4 | 54.3 | 0.601 | 51.7 | 51.7 | 0.608 | 52.2 | 52.2 |
| SVIT | LLaVA-1.5 | 1.000 | 100.0 | 100.0 | 1.000 | 100.0 | 100.0 | 0.979 | 97.5 | 97.5 | 0.861 | 85.0 | 85.0 | 1.000 | 100.0 | 100.0 | 0.863 | 82.5 | 82.5 |
| | MiniGPT-4 | 0.839 | 82.9 | 82.9 | 0.937 | 92.7 | 92.7 | 0.781 | 75.6 | 80.5 | 0.788 | 75.6 | 78.0 | 0.865 | 85.4 | 87.8 | 0.750 | 73.2 | 73.2 |
| | BLIP-2 | 0.776 | 72.7 | 75.8 | 0.840 | 82.9 | 85.4 | 0.611 | 56.5 | 56.5 | 0.550 | 48.5 | 54.3 | 0.596 | 56.1 | 61.0 | 0.504 | 42.3 | 42.3 |
| | InstructBLIP | 0.732 | 71.4 | 74.3 | 0.782 | 74.3 | 74.3 | 0.792 | 73.3 | 73.3 | 0.701 | 65.9 | 65.9 | 0.627 | 55.2 | 55.2 | 0.630 | 53.6 | 53.6 |
| VQAv2 | LLaVA-1.5 | 1.000 | 100.0 | 100.0 | 1.000 | 100.0 | 100.0 | 1.000 | 100.0 | 100.0 | 0.930 | 92.5 | 92.5 | 1.000 | 100.0 | 100.0 | 0.980 | 97.5 | 97.5 |
| | MiniGPT-4 | 0.890 | 87.5 | 87.5 | 0.897 | 87.8 | 92.7 | 0.937 | 92.7 | 92.7 | 0.792 | 75.6 | 78.0 | 0.836 | 80.5 | 87.8 | 0.828 | 80.5 | 80.5 |
| | BLIP-2 | 0.792 | 78.8 | 84.8 | 0.850 | 80.5 | 85.4 | 0.601 | 56.5 | 56.5 | 0.573 | 53.7 | 56.1 | 0.642 | 61.0 | 65.9 | 0.508 | 43.5 | 43.5 |
| | InstructBLIP | 0.755 | 68.8 | 78.1 | 0.800 | 76.7 | 76.7 | 0.695 | 61.5 | 61.5 | 0.634 | 58.5 | 65.9 | 0.688 | 61.0 | 65.9 | 0.576 | 47.4 | 47.4 |

solely conduct targeted attacks, we implement our adversarial transformation-aware attack for comparison (we do not implement the hybrid transformation-based attack as it can only support untargeted attacks). As shown in Table 8 and Table 9, in a fair comparison setting, our attack method also achieves better performance than existing LVLM attacks MF and CroPA. This demonstrates that: (1) A simple and easy-to-implement transformation-aware attack is effective enough to fool the LVLM models. (2) Both MF and CroPA design complicated perturbation patterns. Compared to them, our transformation-aware attack is simple and easy-to-implement with better performance. Overall, we validate that adversarial visual transformation can achieve significant attack performance against LVLM models.

Besides, we also provide the complexity comparison with the two LVLM attacks: MF and CroPA. As shown in Table 10, our transformation attack is much more efficient than previous attackers as they rely on more complicated adversarial pattern designs. Specifically, MF relies on an additional surrogate model CLIP to first initialize the noise and then design a perturbation update process to optimize the noise against the target LVLM, therefore introducing more model memory and time costs. CroPA requires optimizing both visual and textual noise with multi-prompt adversarial train-

Table 8: Performance comparison with the MF attack (Zhao et al., 2024) on the same ImageNet (Deng et al., 2009) dataset with the same semantic similarity metric (↑). The values of the MF attack are reported in its paper.

| Attack | BLIP-2 (Li et al., 2023) | MiniGPT-4 (Zhu et al., 2023) | LLaVA-1.5 Liu et al. (2024a) |
|---|---|---|---|
| Clean image (Zhao et al., 2024) | 0.503 | 0.470 | 0.437 |
| MF-it (Zhao et al., 2024) | 0.546 | 0.484 | 0.452 |
| MF-ii (Zhao et al., 2024) | 0.592 | 0.572 | 0.450 |
| MF-ii+tt (Zhao et al., 2024) | 0.665 | 0.666 | 0.597 |
| Clean image | 0.569 | 0.427 | 0.369 |
| Ours-Adversarial | **0.842** | **0.878** | **0.806** |

Table 9: Performance comparison with the CroPA attack (Luo et al., 2024) on the same MS-COCO (Lin et al., 2014) dataset and OpenFlamingo (Awadalla et al., 2023) model with the same attack success rate metric (↑). The values of the CroPA attack are reported in its paper.

| Attack | $VQA_{general}$ | $VQA_{specific}$ | Classification | Captioning | Overall |
|---|---|---|---|---|---|
| Single-P (Luo et al., 2024) | 0.21 | 0.43 | 0.47 | 0.34 | 0.36 |
| Multi-P (Luo et al., 2024) | 0.60 | 0.85 | 0.71 | 0.60 | 0.69 |
| CroPA (Luo et al., 2024) | 0.90 | 0.96 | 0.75 | 0.72 | 0.83 |
| Ours-Adversarial | **1.00** | **1.00** | **1.00** | **1.00** | **1.00** |

Table 10: Complexity comparison with MF and CroPA attacks.

| Process | Attack Type | GPU Time (↓) | GPU Memory (↓) |
|---|---|---|---|
| Generation | MF (Zhao et al., 2024) | 29min | 35GB |
| | CroPA (Luo et al., 2024) | 14min | 26GB |
| | Ours-Adversarial | **5min** | **22GB** |

ing, also resulting in relatively more time costs. Therefore, it validates that our simple yet efficient adversarial visual transformation is effective enough to fool the LVLM models.

