# OpenReview forum: "Are Large Vision-Language Models Robust to Adversarial Visual Transformations?"
_ICLR.cc/2025/Conference — ICLR 2025 Conference Withdrawn Submission_

### Official Review · Reviewer_3qRx · 2024-10-21

**Soundness:** 3
**Presentation:** 2
**Contribution:** 1
**Rating:** 1
**Confidence:** 4

**Summary:**

This paper investigates the robustness of Vision-Language Models (VLMs) against a set of predefined visual transformations. The authors demonstrate that these transformations significantly degrade the performance of VLMs.

**Strengths:**

This paper is self-explanatory. The authors argue that certain straightforward visual transformations can degrade model performance in a manner similar to adversarial noise, a point that has not been thoroughly explored in prior work. The authors substantiate this hypothesis through extensive experimentation, providing strong empirical evidence.

**Weaknesses:**

A major limitation of this paper lies in its reliance on visually obvious image transformations as proxies for adversarial noise, which presents a conceptual flaw.

In the field of model robustness, we typically distinguish between two types of challenges. The first is robustness to natural image variations, such as blurriness, changes in lighting, or occlusion, which can occur in real-world applications. Models must be resilient to these forms of noise. The second type, adversarial robustness, refers to a model's ability to withstand carefully crafted, imperceptible perturbations that are specifically designed to cause misclassification or incorrect predictions. These adversarial perturbations are subtle but can significantly disrupt model performance, making this area a key focus in recent deep learning research.

The approach taken in this paper applies transformations that drastically alter the appearance of images, neither simulating natural real-world variations nor aligning with the subtlety of typical adversarial attacks. This undermines the relevance and impact of the findings. For example, in Figure 2, the authors present an image so severely distorted that it becomes unrecognizable to both the VLM and human observers. It is not surprising that the model struggles with such transformations—any human would as well.

Overall, the main takeaway of the paper seems to be that "the more extreme the transformation, the less capable a VLM is at recognizing the image." However, this observation is both predictable and trivial. Moreover, it falls outside the conventional boundaries of adversarial robustness studies, where the degree of perturbation is typically constrained to be imperceptible to humans. By ignoring these constraints, the paper fails to meaningfully contribute to the ongoing discourse on VLM security and robustness.

**Questions:**

Unless the authors can identify practical applications for these findings, it is difficult to see them as surprising or particularly insightful. For instance, even slight variations in the mean and standard deviation values used in transformations like `transform.Normalize` can lead to a performance drop in models—let alone the much more pronounced visual transformations applied here. The lack of clear implications or actionable insights leaves the reader questioning the value of these results.

---

### Official Review · Reviewer_6LMy · 2024-10-29

**Soundness:** 2
**Presentation:** 1
**Contribution:** 2
**Rating:** 3
**Confidence:** 3

**Summary:**

This study investigates the resilience of Large Vision-Language Models (LVLMs) to adversarial visual transformations. Unlike prior studies concentrating on perturbations or prompt manipulations, this research scrutinizes the impact of straightforward, easily executable visual transformations such as rotations and flips on LVLMs' performance. The authors carry out an exhaustive evaluation of individual transformations and their combinations, employing both manual and adaptive learning methods to pinpoint the most detrimental transformations. The experiments conducted on various LVLMs underscore the vulnerability of these models to such attacks and underscore the potential for crafting transformation-based attacks that could compromise model performance in both targeted and untargeted scenarios.

**Strengths:**

1. This paper delves into a seldom-explored field by investigating the effects of visual transformations on LVLMs, diverging from traditional adversarial attacks based on perturbations.

2. The authors conduct a comprehensive evaluation of LVLMs' resilience to a broad spectrum of visual transformations, encompassing both spatial and spectral alterations, thereby enriching their analysis.

3. The research utilizes multiple datasets, LVLM models, and rigorous evaluation metrics, providing strong empirical support for the conclusions.

**Weaknesses:**

1. The current study refrains from directly comparing its methodologies with other forms of adversarial attacks. Such a comparison could potentially elucidate the comparative strengths of transformation-based attacks.
2. Lack of discussion about how to defend against them.
3. Overall, this paper seems to be a simple exploration of visual transformations and LVLMs, and does not present new insights or approaches.

**Questions:**

See Weaknesses.
I would love to see further elaboration by the authors to make sure I understand this paper correctly.

---

### Official Review · Reviewer_CQEj · 2024-10-31

**Soundness:** 3
**Presentation:** 3
**Contribution:** 3
**Rating:** 5
**Confidence:** 2

**Summary:**

This paper explores the robustness of Large Vision-Language Models (LVLMs) against adversarial attacks through visual transformations. The authors introduce a novel approach to assess and enhance the adversarial robustness of LVLMs by systematically evaluating the impact of various visual transformations on model performance. They propose a method that combines harmful transformations and employs adversarial learning to dynamically apply these transformations to raw images, aiming to improve the effectiveness and imperceptibility of the attacks. The findings suggest that LVLMs can be significantly misled using well-designed adversarial transformations, highlighting a critical vulnerability in these models.

**Strengths:**

1. **Originality**: The paper introduces a unique perspective on assessing the robustness of LVLMs. It stands out by focusing on the effectiveness of simple visual transformations, rather than complex perturbations, which is a novel direction in adversarial attack studies.
2. **Quality**: The empirical evaluations are thorough and provide compelling evidence of the impact of adversarial transformations on LVLM performance. The use of semantic similarity and attack success rate as metrics is appropriate and allows for clear comparisons.
3. **Clarity**: The manuscript is well-structured and clearly written, making it accessible to a broad audience. The methodologies and findings are presented in a logical sequence that supports the reader's comprehension.
4. **Significance**: The work addresses an important issue regarding the security and reliability of LVLMs, which is becoming increasingly relevant as these models are integrated into real-world applications. The results have the potential to influence the development and deployment strategies of LVLMs.

**Weaknesses:**

1. **Scalability**: While the proposed method demonstrates success in a controlled environment, there is limited discussion on how the adversarial transformations might fare in more complex scenarios, such as real-time applications or dynamic environments.
2. **Generalizability**: The experiments focus primarily on a specific set of LVLMs and tasks. It would be beneficial to see how the adversarial strategies generalize across a wider range of models and tasks.

**Questions:**

- Can the authors elaborate on how the proposed adversarial learning approach could be extended to handle more complex or dynamic visual transformations?
- What measures can be taken to mitigate the potential misuse of the proposed adversarial techniques, especially considering their practical applicability?
- Is there any preliminary research or considerations given to the defense mechanisms that could counteract the described adversarial attacks?

---

### Official Review · Reviewer_QCKr · 2024-11-04

**Soundness:** 2
**Presentation:** 3
**Contribution:** 2
**Rating:** 5
**Confidence:** 4

**Summary:**

This paper sufficiently evaluates the robustness of LVLM to various visual transformations, and proposes a method to generate adversarial examples by combining visual transformations.

**Strengths:**

1. The paper was evaluated on sufficient models (LLaVA-1.5, MiniGPT-4, BLIP-2 and InstructBLIP) and tasks (VQA, Classification, Captioning).

2. This paper is clearly written with intuitive tables and figures.

**Weaknesses:**

1. What is the motivation and the problem to be solved by the thesis? Of the two attack methods proposed by the authors, if the image is attacked using visual transformations, the perturbation of the adversarial example is too large; if the attack is done according to the method in Fig. 3, then why not just attack the image directly through the end-to-end gradient, instead of targeting the visually transformed image?

2. Although the authors justify their lack of comparison with other LVLM attacks in line 513 and give some results in Appendix D. However, I still have doubts that the authors' method compared to end-to-end direct attacks on LVLM, both utilising white-box models and gradients, then the performance of the method should be compared to end-to-end attacks. Obviously, the authors' method will not be better than the end-to-end attack. Despite the authors' claim in line 414 that their method is more practical in a black box setup, I failed to see that the author's method transfer attack in a black box setup yielded better results than an end-to-end black box attack. For example, could the author's method be better than Attack-Bard [1] ? Or, whether the authors' method can attack commercial black-box models？

[1] How Robust is Google's Bard to Adversarial Image Attacks? NeurIPS 2023 Workshop on Robustness of Few-shot and Zero-shot Learning in Foundation Models

3. The selection of datasets. The authors explain the construction of the dataset in line 153, were these samples randomly selected from the dataset or was some manual selection done?

4. In Table 9, the authors compare with CroPA. the goal of CroPA's attack is to make the image output the target word, e.g. None, for many, many prompts. Is the authors' attack in Table 9 also attacked and evaluated on so many prompts? Or is it only attacked and evaluated on a fixed prompt?

**Questions:**

See weaknesses.

---

### Note · Authors · 2024-11-13

I have read and agree with the venue's withdrawal policy on behalf of myself and my co-authors.